# Reproducibility study of "Learning Decision Trees and Forests with Algorithmic Recourse"

## Abstract

Decision trees and random forests are widely recognized machine learning models, particularly for their interpretability. However, ensuring algorithmic recourse—providing individuals with actionable steps to alter model predictions—remains a significant challenge. The authors of the paper **Learning Decision Trees and Forests with Algorithmic Recourse** (Kanamori et al. (2024)) introduce a novel method for training tree-based models while guaranteeing the existence of recourse actions. In this study, we attempt to replicate the original findings and validate their data using the open-source implementation and datasets provided in the original paper. While we observe some differences in the performance of sensitivity forests, we confirm that our results closely align with those of the decision trees presented in the original study.

## 1 Introduction

The paper **Learning Decision Trees and Forests with Algorithmic Recourse** by (Kanamori et al. (2024)) presents a new framework for learning tree-based models that ensures accurate predictions while also guaranteeing the existence of algorithmic recourse actions. Algorithmic recourse (AR) provides affected individuals with actionable steps to alter unfavorable model predictions. This approach aims to bridge the gap between predictive performance and practical usability by ensuring that recourse actions are not only feasible but also reasonable in real-life scenarios. Traditional models often fail to guarantee such recourse, as they prioritize prediction accuracy without considering the cost and feasibility of actionable outcomes.

The authors propose Recourse-Aware Classification Trees (RACT), a novel algorithm that incorporates adversarial training techniques to learn decision trees and random forests capable of guaranteeing recourse. By extending a standard top-down greedy approach (similar to CART), their algorithm balances predictive performance with the existence of valid actions for more instances. The method is further extended to random forests, making it applicable to a broader range of real-world problems.

As part of our university course, we attempted to reproduce the results presented in the paper. Our objective was to evaluate the validity of the proposed RACT algorithm. We carefully followed the implementation details described by the authors, applied the algorithm to the same real-world datasets the authors used, and compared the outcomes with the original findings. This report outlines our efforts in reconstructing the experiments, highlighting the challenges faced, the insights gained, and the extent to which the original results were replicated successfully.

## 2 Reconstruction and Validation Process

To ensure the reliability and reproducibility of our results, we closely followed the implementation details provided in the paper and utilized the code available in the authors' GitHub repository. However, upon initial testing, we observed inconsistent and differing results across our runs.

Our first step was to examine the package versions used in the authors' experiments. The repository and paper did not provide sufficient details on this, so we reached out to the authors for clarification. This

was crucial, as mismatched library versions can lead to subtle yet significant variations in performance and accuracy. The authors kindly shared the exact package versions, enabling us to align our environment with theirs and establish a more consistent baseline for comparison. We then tested the algorithm on multiple systems with varying hardware configurations, all running Windows Subsystem for Linux (WSL), to validate its behavior further. Additionally, we leveraged the high-performance computing (HPC) system we had access to, to assess the algorithm's scalability and performance under optimal computational conditions. Although the original authors used macOS, we considered this difference unlikely to significantly affect the results.

After running the experiments with the same settings and package versions on our different setups, we observed discrepancies in parts of the outcomes. The provided code seemed to include stochastic elements that were not controlled by a fixed seed. Upon closer inspection, we suspected these inconsistencies were due to non-thread-safe behavior in the code.

To investigate further, we focused on verifying whether the authors' multi-threading implementation was truly thread-safe. Improper multi-threading management can lead to race conditions and inconsistent results. We compared the outcomes of both single-threaded and multi-threaded runs across different systems to identify any concurrency or thread-safety issues. When we found that single-threaded runs were consistent, while multi-threaded runs produced varying results, we concluded that the provided code was not thread-safe.

The authors acknowledged that they had not specifically focused on ensuring thread-safe behavior. This prompted us to investigate further to determine whether the results presented by the authors were drawn from the same distribution. To address this, we attempted to conduct a statistical test to assess the stochastic nature of the results.

These extensive tests, conducted across various platforms and configurations, provided valuable insights into the stability and scalability of the proposed method. Although our results differed from those of the authors, we were able to gain a deeper understanding of the algorithm's behavior and its potential limitations. Ultimately, our findings highlight areas for improvement, particularly regarding thread safety, and contribute to a more robust evaluation of the algorithm's practical applicability.

## 3   Scope of Reproducibility

The study by Kanamori et al. introduces a novel approach for learning decision trees and forests while ensuring Algorithmic Recourse (AR). The authors propose the Recourse-Aware Classification Tree (RACT) framework, which aims to provide affected individuals with actionable steps to alter an undesirable classification outcome. The method is designed to balance predictive accuracy, recourse guarantees, and computational efficiency, addressing a key challenge in algorithmic fairness.

In this study, we seek to verify the key claims made in the original paper by attempting to reproduce its experiments. Specifically, we assess the following hypotheses:

1. RACT successfully provides recourse actions to more individuals than baseline methods (Vanilla, OAF) while maintaining comparable predictive accuracy.

2. The proposed learning algorithm is computationally efficient, meaning that its runtime is comparable to traditional decision tree training methods.

3. RACT enables controlling the trade-off between predictive accuracy and recourse guarantees by adjusting a so-called trade-off parameter $\lambda$.

4. The learned models produce low-cost and plausible recourse actions, ensuring that the suggested changes to feature values are realistic and executable in practice.

By rigorously evaluating these claims, we aim to assess the reliability, generalizability, and practical implications of the RACT framework in real-world applications.

# 4 Method

In this section, we outline the experimental setup used to reproduce the results of the original study on Recourse-Aware Classification Trees (RACT). We closely followed the methodology described by the authors, utilizing their provided datasets and implementation to ensure a faithful reproduction of their findings.

## 4.1 Datasets

The original study was conducted on multiple real-world datasets, which were made available by the authors in their repository. For consistency, we used the exact same datasets without modifications.

**FICO** (N = 9871, D = 23) – A credit risk dataset used for predicting loan approvals.

**COMPAS** (N = 6167, D = 14) – A dataset containing criminal justice risk assessment scores.

**Credit** (N = 30,000, D = 16) – A dataset related to consumer credit risk classification.

**Bail** (N = 8923, D = 16) – A dataset used for predicting bail outcomes.

The datasets were provided in the GitHub repository and used as-is, ensuring consistency with the original study.

## 4.2 Experimental Setup

We executed the experiments using the official code provided by the authors, ensuring that all steps, including model training, evaluation, and comparative analyses, remained unchanged. The study focused on the Recourse-Aware Classification Tree (RACT), a decision tree classifier designed to ensure algorithmic recourse, as well as its Random Forest extension. They consistently compared their approach with own implementations of CART (later referred to as Vanilla) and Only-Actionable-Features (OAF), as introduced in Dominguez-Olmedo et al. (2022).

To maintain comparability, we followed the same experimental design as the original study:

- Cross-Validation: A 10-fold cross-validation procedure was applied across all models, including RACT and its Random Forest version, ensuring robustness and minimizing variance in performance estimation.

- Performance Analysis: The models were evaluated on standard classification metrics, comparing RACT to baseline methods.

- Recourse Analysis: We examined the quality and availability of recourse actions generated by RACT, assessing whether individuals received feasible alternatives to change an undesirable classification outcome.

- Computational Efficiency: We analyzed the runtime of RACT compared to traditional decision trees, verifying the authors' claims regarding efficiency.

- Trade-off Parameter Analysis: The study explored the impact of the trade-off parameter $\lambda$, which balances predictive accuracy and recourse guarantees.

## 4.3 Hardware and Software Environment

To ensure reproducibility, we conducted the experiments in an environment closely aligned with the original study. The experiments were performed on both the HPC cluster and a local machine equipped with an AMD Ryzen 5 7600 processor. While the original study was conducted on macOS, we executed our reproduction in a Linux environment on the HPC and in Windows 11 using the Windows Subsystem for Linux (WSL) with Ubuntu. Although these setups differ, we expect them to have minimal impact on the results, except for possibly runtime performance. The specific libraries and tools utilized in the implementation of the experiments are listed in the appendix.

### 4.4 Reproducibility Considerations

Since we directly utilized the authors' code and datasets, the reproduction focused on verifying the consistency of the reported results rather than reimplementing the method. Any deviations from the expected results were carefully examined to determine whether they stemmed from implementation discrepancies, hardware differences, or statistical variation inherent in the cross-validation procedure.

By adhering to this methodology, we aimed to rigorously validate the claims of the original study and assess the generalizability of the RACT framework in real-world applications.

### 4.5 Model Evaluation and Statistical Analysis

Parts of the code could be directly compared, particularly the sections that relied solely on the tree classifier. These parts were thread-safe (based on our findings), meaning we could simply compare them. However, we discovered that some parts of the code had stochastic behavior when running in multithreaded environments, which led us to believe that the original authors' code might also be stochastic. This meant we couldn't just do a direct comparison, so we needed to use a different method. Specifically, we performed statistical testing to determine whether their results and ours came from the same probability distribution.

To improve the accuracy of our statistical analysis, we took advantage of the multiple values generated by the 10-fold cross-validation (CV). Each of the final results shown in the authors' plots was the mean of these 10 values, so we used this information for our analysis.

Here's the methodology we used:

- Null Hypothesis $H_0$: We first set up the null hypothesis that the results of our experiment are drawn from the same probability distribution than the results of the authors.

- 10-Fold Cross-Validation (CV) Results: For each given $\lambda$-value, we had ten results from the 10-fold CV. We compared the individual results (rather than just the mean values) from both our and the authors' experiments.

- Alternative Hypothesis $H_0'$: We then formulated a new hypothesis: $H_0'$ – that 50% of the time, our result is higher than theirs. We assumed that if $H_0$ holds true, it is implied that $H_0'$ should also be valid. In other words, if $H_0'$ is unlikely, we can also reject $H_0$.

- Comparison of Results: We compared the corresponding values from our runs and the authors' runs, counting how often our values were larger than theirs and vice versa.

- Statistical Analysis: Using a two-sided binomial test with parameters n = 10 and p = 0.5, we calculated p-values to assess the probability that the observed differences could have arisen by chance.

## 5 Results

Our efforts to replicate the experiments from the original study revealed a number of insights into the behavior of the Recourse-Aware Classification Tree (RACT) and its Random Forest extension.

### 5.1 RACT Tree Classifier

The RACT tree classifier, which forms the core of the proposed method, was fully reproducible in our experiments (see e.g. Figure 1). The results we obtained for the decision tree model closely aligned with those reported by the original authors. This included the key findings related to the provision of algorithmic recourse and the balance between prediction accuracy and feasibility of recourse actions.

Our reproduction of the decision tree experiments confirmed that RACT successfully provides recourse actions to a greater number of individuals compared to baseline methods (Vanilla and OAF), while maintaining

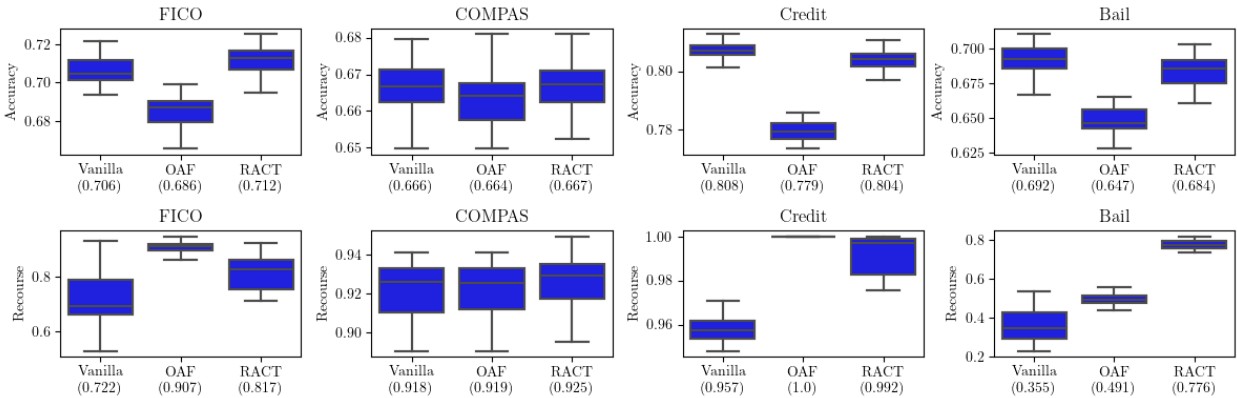

Figure 1: RACT classification tree in comparison with baselines, identical in this case. Reproduction and expansion of Figure 3 a) in Kanamori et al. (2024)

| Dataset | Vanilla | OAF | RACT |
|---|---|---|---|
| FICO | $0.36 \pm 0.08$ | $0.272 \pm 0.02$ | $0.408 \pm 0.07$ |
| COMPAS | $0.038 \pm 0.01$ | $0.03 \pm 0.0$ | $0.038 \pm 0.0$ |
| Credit | $1.758 \pm 0.2$ | $1.27 \pm 0.15$ | $1.669 \pm 0.21$ |
| Bail | $0.188 \pm 0.02$ | $0.055 \pm 0.01$ | $0.188 \pm 0.03$ |
| Dataset | Vanilla | OAF | RACT |
| FICO | $42.085 \pm 1.8$ | $35.25 \pm 1.52$ | $48.074 \pm 2.33$ |
| COMPAS | $2.311 \pm 0.32$ | $1.755 \pm 0.14$ | $2.074 \pm 0.13$ |
| Credit | $292.532 \pm 21.56$ | $268.405 \pm 18.93$ | $277.398 \pm 25.75$ |
| Bail | $24.842 \pm 1.93$ | $3.262 \pm 0.35$ | $22.968 \pm 1.2$ |

Table 1: Results on the average running time in [s] for the random forest classifiers (our results on top). Reproduction and expansion of Table 1 in Kanamori et al. (2024)

comparable predictive accuracy. Furthermore, the analysis of computational efficiency supported the authors' claims regarding the efficiency of the RACT algorithm, with runtimes similar to those of traditional decision tree classifiers.

## 5.2 RACT Random Forest Classifier

In contrast to the RACT tree classifier, the Random Forest extension exhibited significant inconsistencies (see Figure 2 or Figure 3 or Tables 1, 2, 3, 4. We traced these discrepancies in parts to a thread-safety issue in the original code, which was not explicitly addressed by the authors. Specifically, we found that the multi-threaded implementation of the Random Forest algorithm produced varying results on different runs, while single-threaded runs yielded consistent outcomes.

Most of the figures presented in the original paper were generated using the Random Forest classifier, and as such, our reproduction efforts showed significant deviations, particularly in the sensitivity analysis. The differences were statistically significant, as confirmed by a binomial test comparing our results with those of the original study. This is a critical finding, as the authors' results were heavily reliant on the forest-based models.

## 5.3 Random Forests

In contrast, the results for the Random Forest extension of RACT were less consistent.

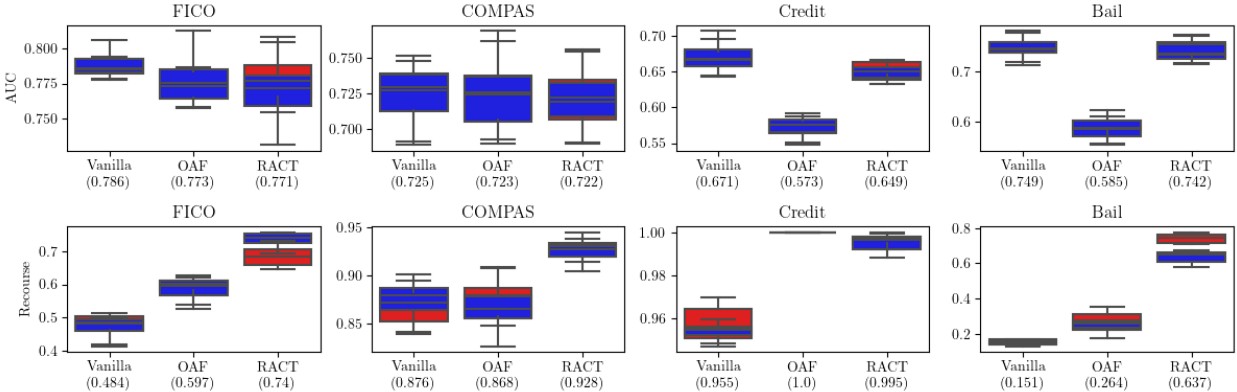

Figure 2: RACT Random forests in comparison with baselines, ours in red, authors in blue. Reproduction and expansion of Figure 3 b) in Kanamori et al. (2024)

| Dataset | Vanilla | OAF | RACT |
|---------|---------|-----|------|
| FICO | $0.366 \pm 0.11$ | $0.159 \pm 0.02$ | $\mathbf{0.28 \pm 0.08}$ |
| COMPAS | $0.182 \pm 0.02$ | $0.179 \pm 0.02$ | $\mathbf{0.176 \pm 0.01}$ |
| Credit | $0.253 \pm 0.02$ | $0.001 \pm 0.0$ | $\mathbf{0.163 \pm 0.04}$ |
| Bail | $0.581 \pm 0.09$ | $0.419 \pm 0.02$ | $\mathbf{0.22 \pm 0.01}$ |
| Dataset | Vanilla | OAF | RACT |
| FICO | $0.447 \pm 0.05$ | $0.407 \pm 0.03$ | $\mathbf{0.283 \pm 0.01}$ |
| COMPAS | $0.298 \pm 0.02$ | $0.28 \pm 0.01$ | $\mathbf{0.232 \pm 0.02}$ |
| Credit | $0.293 \pm 0.02$ | $0.0 \pm 0.0$ | $\mathbf{0.166 \pm 0.04}$ |
| Bail | $0.763 \pm 0.03$ | $0.525 \pm 0.04$ | $\mathbf{0.332 \pm 0.04}$ |

Table 2: Valid costs (our results on top). Reproduction and expansion of Table 2 in Kanamori et al. (2024)

| Dataset | Vanilla | OAF | RACT |
|---------|---------|-----|------|
| FICO | $0.48 \pm 0.01$ | $0.459 \pm 0.0$ | $\mathbf{0.479 \pm 0.01}$ |
| COMPAS | $0.452 \pm 0.01$ | $0.453 \pm 0.01$ | $\mathbf{0.451 \pm 0.01}$ |
| Credit | $0.521 \pm 0.01$ | $0.213 \pm 0.3$ | $\mathbf{0.509 \pm 0.01}$ |
| Bail | $\mathbf{0.513 \pm 0.01}$ | $0.5 \pm 0.0$ | $0.515 \pm 0.0$ |
| Dataset | Vanilla | OAF | RACT |
| FICO | $0.456 \pm 0.0$ | $0.446 \pm 0.0$ | $\mathbf{0.437 \pm 0.0}$ |
| COMPAS | $\mathbf{0.44 \pm 0.01}$ | $0.447 \pm 0.01$ | $0.453 \pm 0.01$ |
| Credit | $0.526 \pm 0.01$ | $0.0 \pm 0.0$ | $\mathbf{0.523 \pm 0.01}$ |
| Bail | $\mathbf{0.504 \pm 0.01}$ | $0.507 \pm 0.0$ | $0.512 \pm 0.01$ |

Table 3: Plausibility (our results on top). Reproduction and expansion of Table 3 in Kanamori et al. (2024)

| Dataset | Vanilla | OAF | RACT |
|---------|---------|-----|------|
| FICO | $0.487 \pm 0.16$ | $0.822 \pm 0.04$ | $\mathbf{0.619 \pm 0.13}$ |
| COMPAS | $0.745 \pm 0.05$ | $0.749 \pm 0.05$ | $\mathbf{0.762 \pm 0.05}$ |
| Credit | $0.542 \pm 0.08$ | $1.0 \pm 0.0$ | $\mathbf{0.838 \pm 0.09}$ |
| Bail | $0.212 \pm 0.09$ | $0.413 \pm 0.03$ | $\mathbf{0.722 \pm 0.03}$ |
| Dataset | Vanilla | OAF | RACT |
| FICO | $0.316 \pm 0.04$ | $0.502 \pm 0.03$ | $\mathbf{0.648 \pm 0.03}$ |
| COMPAS | $0.599 \pm 0.02$ | $0.642 \pm 0.04$ | $\mathbf{0.721 \pm 0.02}$ |
| Credit | $0.669 \pm 0.05$ | $1.0 \pm 0.0$ | $\mathbf{0.964 \pm 0.03}$ |
| Bail | $0.174 \pm 0.04$ | $0.201 \pm 0.04$ | $\mathbf{0.649 \pm 0.03}$ |

Table 4: Budget validity (our results on top). Reproduction and expansion of Table 4 in Kanamori et al. (2024)

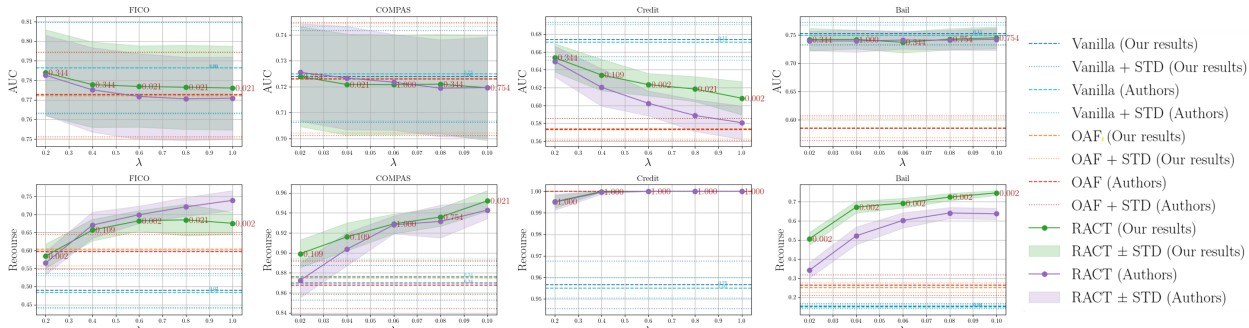

Figure 3: Sensitivity analysis with respect to the trade-off parameter $\lambda$. Reproduction and expansion of figure 4 in Kanamori et al. (2024)

## 5.4 Statistical Testing and Sensitivity Analysis

Because of the inconsistencies with the Random Forest classifier, we needed a different method to compare the results of our experiments with the results of the authors. We were able to perform a statistical analysis using the 10-fold cross-validation results from both our runs and the authors'. Our analysis showed that the observed differences between the results of our experiments and those of the original study could not be attributed solely to chance, as indicated by the significant p-values from the binomial test. This was particularly evident in the sensitivity analysis (see Figure 3, where our results differed notably from the original findings.

It is worth noting that, in cases where we observed differing results, our implementation often outperformed the original results in terms of both predictive accuracy and recourse availability. While this suggests that the RACT framework holds promise for real-world applications, it also underscores the potential for further improvements to the implementation, particularly in the Random Forest version.

## 6 Challenges and Resolutions

Throughout our replication process, we encountered several practical challenges that required careful problem-solving. One major issue was that the code repository did not include a list of required packages or their specific versions, making it difficult to recreate the original environment. To overcome this, we reached out directly to the authors, who were very helpful in providing a complete pip freeze output with all the necessary package details. Another challenge arose from the hardware differences: the original experiments were conducted on a Mac running macOS Monterey with an Apple M1 Pro CPU and 32 GB of memory—a configuration we did not have access to. As a workaround, we employed Windows Subsystem for Linux (WSL) and leveraged the HPC , which operates on Linux, to run the experiments reliably. Lastly,

we faced a discrepancy regarding the depth of trees and forests. The paper mentioned a maximum tree depth of 64, but the provided code used for the forest classifier much shallower maximum depths at 16. We experimented with both settings and found that the full depth of 64 was achievable only on the HPC, which used 128 GB of RAM (using only 64GB of memory lead to memory overflow), leading us to suspect that the depth mentioned in the paper might be a typographical error. This suspicion is reinforced by the impracticality of running such deep trees in the forest classifier setting with 200 trees. Overall, these challenges prompted us to adapt our approach, ensuring that our replication remained robust and our results reliable despite the initial hurdles.

## 7 Conclusion

Despite addressing most of the identified issues, our experiments consistently produced results that deviated from those reported in the original study. In fact, our outcomes were not only comparable to but in many cases even better than the published results. We ran the original code and directly compared the outputs across a wide range of hardware platforms. Interestingly, despite the different computing environments, we consistently obtained similar results on our different hardware, which underscores the robustness of our findings. However, they did not match the results presented in the paper.

Although our replication did not exactly mirror the reported outcomes, the fact that our approach often delivered superior performance is encouraging. It demonstrates that their implementation is both reliable and efficient, opening up promising avenues for further improvements and refinements in their proposed framework.

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

## A Appendix

### A.1 Python packages:

| Paket | Version |
|---|---|
| aif360 | 0.5.0 |
| aiohttp | 3.9.5 |
| aiosignal | 1.3.1 |
| alembic | 1.13.2 |
| altair | 5.3.0 |
| anyio | 3.6.2 |
| appnope | 0.1.3 |
| apricot-select | 0.6.1 |
| argon2-cffi | 21.3.0 |
| argon2-cffi-bindings | 21.2.0 |
| arrow | 1.2.3 |
| arviz | 0.18.0 |
| asttokens | 2.2.1 |
| async-timeout | 4.0.3 |
| attrs | 22.2.0 |
| backcall | 0.2.0 |

| | |
|---|---|
| beautifulsoup4 | 4.11.2 |
| bleach | 6.0.0 |
| blinker | 1.7.0 |
| cachetools | 5.3.3 |
| cdt | 0.6.0 |
| certifi | 2023.5.7 |
| cffi | 1.15.1 |
| cfgv | 3.3.1 |
| charset-normalizer | 3.1.0 |
| click | 8.1.7 |
| cloudpickle | 3.0.0 |
| colorlog | 6.8.2 |
| comm | 0.1.2 |
| cons | 0.4.6 |
| contourpy | 1.0.7 |
| cycler | 0.11.0 |
| Cython | 0.29.35 |
| debugpy | 1.6.6 |
| decorator | 5.1.1 |
| defusedxml | 0.7.1 |
| distlib | 0.3.6 |
| dm-tree | 0.1.8 |
| etuples | 0.3.9 |
| exceptiongroup | 1.2.0 |
| executing | 1.2.0 |
| factor-analyzer | 0.4.1 |
| fastjsonschema | 2.16.3 |
| filelock | 3.10.2 |
| folktables | 0.0.12 |
| fonttools | 4.38.0 |
| fqdn | 1.5.1 |
| frozenlist | 1.4.1 |
| fsspec | 2024.6.0 |
| future | 0.18.3 |
| gitdb | 4.0.11 |
| GitPython | 3.1.43 |
| GPUtil | 1.4.0 |
| GPy | 1.13.2 |
| GraKeL | 0.1.9 |
| graphviz | 0.20.1 |
| h5netcdf | 1.3.0 |
| h5py | 3.11.0 |
| identify | 2.5.21 |
| idna | 3.4 |
| igraph | 0.10.4 |
| importlib-metadata | 6.11.0 |
| iniconfig | 2.0.0 |
| ipykernel | 6.21.2 |
| ipython | 8.11.0 |
| ipywidgets | 8.0.4 |
| isoduration | 20.11.0 |
| jedi | 0.18.2 |
| Jinja2 | 3.1.2 |
| joblib | 1.2.0 |

| | |
|---|---|
| jsonpointer | 2.3 |
| jsonschema | 4.17.3 |
| jupyter | 1.0.0 |
| jupyter-console | 6.6.2 |
| jupyter$_c$lient | 8.0.3 |
| jupyter$_c$ore | 5.2.0 |
| kiwisolver | 1.4.4 |
| lightgbm | 3.3.5 |
| lightning-utilities | 0.11.2 |
| lingam | 1.9.0 |
| llvmlite | 0.39.1 |
| Mako | 1.3.5 |
| matplotlib | 3.7.0 |
| networkx | 3.0 |
| numba | 0.56.4 |
| numpy | 1.23.4 |
| pandas | 1.5.3 |
| plotly | 5.20.0 |
| protobuf | 4.25.3 |
| pytorch-lightning | 2.2.5 |
| scikit-learn | 1.5.2 |
| scipy | 1.10.1 |
| seaborn | 0.12.2 |
| shap | 0.44.0 |
| SQLAlchemy | 2.0.31 |
| statsmodels | 0.13.5 |
| streamlit | 1.39.0 |
| sympy | 1.12 |
| torch | 2.0.1 |
| xgboost | 1.7.4 |
| zipp | 3.18.1 |

