# OpenReview forum: "Reproducibility study of "Learning Decision Trees and Forests with Algorithmic Recourse""
_TMLR — Rejected by TMLR_

### Review · Reviewer_ku9W · 2025-03-11

**Summary Of Contributions:**

This paper conducts a reproducibility study of the paper "Learning Decision Trees and Forests with Algorithmic Recourse [Kanamori+ 24]," which was published in ICML2024.
The authors replicate the experiments of the original paper using published source code and, through careful evaluation, point out the thread-safety issues in the original implementation.
While they observed somewhat different results from those reported by the original paper, the authors confirmed that the proposed method in [Kanamori+ 24] often could deliver superior performance compared to the baseline methods.

**Audience:**

Yes

**Claims And Evidence:**

Yes

**Requested Changes:**

- Could you explicitly identify which specific part of the original algorithm or implementation was responsible for the thread-safety issue? In addition, please discuss how generalizable this issue may be to other studies that interest the TMLR audience. I believe the thread-safety issue pointed out in this paper can serve as a valuable practical lesson if it is generalizable to some extent.
- Please consider adding some references that explain the background of algorithmic recourse (e.g., [Ustun+ 19] and [Karimi+ 22]). I think citing related papers helps readers understand specific evaluation criteria in the experiments, such as validity and cost.

[Ustun+ 19] Berk Ustun, Alexander Spangher, Yang Liu. Actionable Recourse in Linear Classification. FAT*, 2019.

[Karimi+ 22] Amir-Hossein Karimi, Gilles Barthe, Bernhard Schölkopf, Isabel Valera. A Survey of Algorithmic Recourse: Contrastive Explanations and Consequential Recommendations. ACM Computing Surveys, 2022.

**Strengths And Weaknesses:**

Strengths:
- This paper is well-written and easy to follow. The presentation of the experimental design and results is clear.
- Through careful replication and evaluation, the authors point out the thread-safety issues in the original implementation of [Kanamori+ 24].

Weaknesses:
- One of my concerns is that it is not identified which specific part of the original algorithm or implementation was responsible for the thread-safety issues. It looks unclear whether this issue can occur in other studies in general or is specific to the RACT algorithm. I am not sure how generalizable the insights provided by this paper are to the other areas of interest to the TMLR audience.
- Another concern is that the scientific insights provided by this paper seem unclear. The authors state that they were able to gain a deeper understanding of the algorithm's behavior and potential limitations. However, I could only find descriptions of its implementation and environments, not its algorithmic details.

---

### Review · Reviewer_gyuL · 2025-03-11

**Summary Of Contributions:**

In this paper, the authors study the reproducibility of _"Learning Decision Trees and Forests with Algorithmic Recourse"_ (Kanamori et al., 2024). They replicate the experimental results presented in the main body of the original paper (excluding those in the appendix) using the published source code and datasets, while addressing discrepancies in software versions and computational environments.

They identify a discrepancy between their reproduced results and a subset of the results reported in the original paper. Through statistical testing, they demonstrate that this difference is not due to random variation but rather to the something related to the code given to reproduce the results, that the authors guess to be the thread-unsafe implementation of the multi-threaded code used in the original study. Additionally, they note a difference in tree depth settings between the paper and the provided code, requiring substantial computational resources to match the reported configuration. Their findings confirm that:

(i) The results of the original paper that do not involve parallelized code execution can be successfully reproduced.

(ii) Although some results (specifically those generated with the multi-threaded code of the original paper) cannot be fully replicated, the new results remain consistent with the key claims of the original paper.

**Audience:**

No

**Broader Impact Concerns:**

No broader impact concerns.

**Claims And Evidence:**

No

**Requested Changes:**

# Critical changes

- The authors should expand their study by either providing insights into the generalizability of the results or identifying contradictions beyond merely confirming that the shared code produces results that support the claims in the original paper.
- The authors should provide a more detailed explanation of the statistical test they used, including a justification for their choice over alternative methods (such as the Mann–Whitney U test) and a description of how they preprocessed the experimental results before applying the test. Additionally, they should report the obtained p-values and discuss their implications.
- The authors should improve the presentation by removing redundant content and providing a clearer comments on their findings, including figures, tables, and the statistical test results.

(see also the _"Strengths And Weaknesses Section"_ of the review)

# Changes that are not critical but can strengthen the work

- Add background information.
- Fix the typos and enlarge the figures.

**Strengths And Weaknesses:**

Thank you to the authors for submitting this paper! Assessing the reproducibility of the huge number of papers in the AI field is important without any doubts, and I appreciate the purpose of this study.

Even though the authors have done a good job conveying the challenges they faced in reproducing the results, I think that the current paper needs improvements in both content and presentation before being acceptable at TMLR. Below, I outline the strengths and weaknesses of the paper and hope the authors will take these comments into consideration when revising their work.

# Strengths
The authors explained and detailed the challenges that tthey encountered while attempting to reproduce the results, such as the absence of specific package versions, that is appreciable. Moreover, they successfully replicate some of the key results from the main body of the original paper and offer a reasonable hypothesis regarding the irreproducibility of the results obtained by using the multi-threaded code of the original paper. The author's claim about the truth of the claims of the original paper is sound, since the new results obtained in this paper support the claims of the original paper.

# Weaknesses

## Content

The main weakness of this work is that it only tries to reproduce the results of (the main body of) the paper, without trying to propose new theoretical or empirical findings regarding the proposal of the original paper or some contradictions with respect to the claims of the original work. The TMLR acceptance criteria web page explicitly state that ``report that re-runs the experiments of a published paper has educational value to the students involved. But if it doesn't surface generalizable insights, it is unlikely to be of interest to (even a subset of) the TMLR audience``. Thus, the authors should expand their study by either providing insights into the generalizability of the results or identifying contradictions beyond confirming that the shared code produces results that support the claims in the original paper.

Second, I have some concerns regarding the use of statistical testing to determine whether the discrepancy between the reproduced results and those in the original paper is due to chance. First, it is not clear the rationale behind using the binomial test. This test is typically used when the target variable is categorical with two possible outcomes, whereas in this study, the authors are comparing distributions of continuous numerical data (obtained by reproducing the results of the original paper). How are the authors preprocessing the data before applying this test? If the goal is to assess whether two data groups come from the same probability distribution, a more appropriate choice might be the Mann–Whitney U test.  Second, the p-values obtained from the binomial test are not shown in Section 5.4 but they are commented in the text. The authors should include in the paper to substantiate their claims.

## Presentation
The presentation of the paper can certainly be improved. Some concepts are repeated across different sections, such as ``The study by Kanamori et al. introduces a novel approach for learning decision trees and forests while ensuring Algorithmic Recourse (AR).`` (Section 3, already mentioned in Section 1), the fact that the authors use the same implementation of the original paper (Section 4, already mentioned in Section 2) and the information about the used OS and machine to reproduce the results (Section 4.3, already mentioned in Section 2).

Moreover, Section 5, that presents the experimental results, lacks sufficient comments and explicit references to figures and tables. This makes it difficult for readers to connect the textual discussion with the reported data. The authors should explicitly refer to specific figures and tables in the text and integrate key results into the discussion to improve readability.

Finally, the authors should consider to add some additional background information to make the paper more self-contained. A brief introduction to decision trees, random forests, algorithmic recourse, and the meaning of the metrics used in the reproduced experiments would provide helpful context, especially for readers unfamiliar with the original study.

### Minor

The authors should carefully proofread their paper to fix typos, such as:
- missing parenthesis in ``Tables 1,2,3,4`` (Section 5.2);
- missing parenthesis in ``the original findings`` (Section 5.4);

Finally, the figures in Figure 3 are difficult to read. Please, enlarge them.

---

### Review · Reviewer_onzt · 2025-04-02

**Summary Of Contributions:**

The paper is literally a university course project report on validating the results of the paper "Learning Decision Trees and Forests with Algorithmic Recourse". There is no novelty, no contribution or new implementation. It is quite simply an exercise in running the authors code and accounting for discrepancies.

**Audience:**

No

**Claims And Evidence:**

Yes

**Requested Changes:**

N/A

**Strengths And Weaknesses:**

It is a reasonably well written white paper. However, there is no novelty and therefore I don't think this work needs peer review (because the paper itself is a peer-review, just very deep).

---

### Decision · Action_Editor_3Wjv · 2025-05-01

**Recommendation:** Reject

**Comment:**

All the reviewers raised concerns regarding the novelty of this paper.
The paper only reproduces the main content of the original work without attempting to provide insights into the generalizability of the original results or identify any contradictions.
There were no responses from the authors, and the contribution of this paper remains questionable.

**Audience:**

Because the paper does not delivery any deeper insights beyond the original work, it would not be of interest to the TMLR audience.

**Claims And Evidence:**

The paper reproduces the main content of the original work.
The reported results will be correct.